# Suitability of the Nisin Z-producer *Lactococcus lactis* subsp. *lactis* CBM 21 to be Used as an Adjunct Culture for Squacquerone Cheese Production

**DOI:** 10.3390/ani10050782

**Published:** 2020-04-30

**Authors:** Lorenzo Siroli, Francesca Patrignani, Margherita D’Alessandro, Elisa Salvetti, Sandra Torriani, Rosalba Lanciotti

**Affiliations:** 1Department of Agricultural and Food Sciences, University of Bologna, Piazza Goidanich 60, 47521 Cesena, Italy; lorenzo.siroli2@unibo.it (L.S.); margheri.dalessandr3@unibo.it (M.D.); rosalba.lanciotti@unibo.it (R.L.); 2Interdepartmental Center for Industrial Agri-food Research, University of Bologna, Piazza Goidanich 60, 47521 Cesena, Italy; 3Department of Biotechnology, University of Verona, Strada le Grazie 15, 37134 Verona, Italy; elisa.salvetti@univr.it (E.S.); sandra.torriani@univr.it (S.T.)

**Keywords:** Squacquerone cheese, *Lactococcus**lactis* subsp. *lactis*, proteolytic profile, aroma molecule profile, texture features

## Abstract

**Simple Summary:**

This study investigates the potential of a nisin producer, i.e., *Lactococcus*
*lactis* strain, in the making of Squacquerone cheese. The finding of this research indicates that the tested *Lactococcus lactis* strain represents a suitable candidate to be used as adjunct culture in Squacquerone cheesemaking since it improved the safety and sensory quality of the product without negatively affecting rheological characteristics and proteolysis.

**Abstract:**

This research investigated the technological and safety effects of the nisin Z producer *Lactococcus lactis* subsp. *lactis* CBM 21, tested as an adjunct culture for the making of Squacquerone cheese in a pilot-scale plant. The biocontrol agent remained at a high level throughout the cheese refrigerated storage, without having a negative influence on the viability of the conventional *Streptococcus thermophilus* starter. The inclusion of CBM 21 in Squacquerone cheesemaking proved to be more effective compared to the traditional one, to reduce total coliforms and *Pseudomonas* spp. Moreover, the novel/innovative adjunct culture tested did not negatively modify the proteolytic patterns of Squacquerone cheese, but it gave rise to products with specific volatile and texture profiles. The cheese produced with CBM 21 was more appreciated by the panelists with respect to the traditional one.

## 1. Introduction

The launch on the market of new foods focused on the changing consumer needs and aimed to increase food safety and shelf-life is surely a pivotal challenge for the food industry, especially the dairy industry. In fact, innovative technologies able to increase the sustainability of the production processes (through the better exploitation of the raw materials), are important tools to open new business opportunities (providing benefits to consumers) [1,2]. Consequently, the interest in selected microbial strains able to increase cheese safety and shelf-life as well as to diversify the cheese sensory profile has markedly increased [3,4,5]. Interesting results have been obtained for several foods, including certain dairy products, by the use of an adjunct culture of *Lactococcus lactis* strains able to produce nisin [6,7,8,9]. Nisin was the first characterized bacteriocin and its use is permitted in some applications as a food preservative in the European Union [10,11,12,13]. Nisin, mainly the Z type, is characterized, both in vivo and in vitro, by strong antimicrobial activity against spoiling and pathogenic Gram-positive bacteria, associated with great stability and solubility in different food matrices [14,15,16,17,18,19,20,21]. In addition, the use of nisin in combination with chemical–physical treatments, able to modify the cell wall permeability and consequently increase the interaction of nisin with the cytoplasmic membrane, can extend the nisin effectiveness to Gram-negative microorganisms including *Pseudomonas* spp., *Escherichia coli,* and *Salmonella* spp. [22].

The ability of *L. lactis* to produce nisin, together with many other antimicrobial compounds including hydrogen peroxide, CO_2_, organic acids, diacetyl, and other bacteriocins under several environmental conditions further enhances the high potential of this species of lactic acid bacteria for cheese quality improvement and innovation. In fact, *L. lactis* is reported to play a pivotal role, not only in product safety and shelf-life enhancement, but also in rapid milk acidification and in the sensory properties of several dairy products [23,24]. In fact, this species is reported to widely positively affect the aroma and the texture profiles (including moisture content, cohesiveness, softness) of dairy products, directly by its proteolytic and amino acid conversion activities [25], and the creation of optimal biochemical conditions during ripening [26]. Furthermore, *L. lactis* is widely recognized to play a key role during cheese making and early ripening. However, in late ripening, *L. lactis* is generally replaced mainly by lactobacilli, and some authors highlighted the presence of metabolically active cells of *L. lactis* in late ripening stages [27,28,29,30,31,32,33]. In addition, *L. lactis* has extensive use in milk fermentation both in smaller scale traditional productions and in larger scale industrial applications [26,34]. Indeed, *L. lactis* strains, selected on the basis of their technological characteristics, are the main components of dairy starters for the manufacturing of cheese, sour cream, and buttermilk [35,36,37,38].

In this context, the aim of this research was to evaluate the potential of *L. lactis* subsp. *lactis* CBM 21 (a previously characterized nisin Z-producer strain of dairy origin; [13]) in the diversification and improvement of quality, safety, and shelf-life of Squacquerone cheese.

Squacquerone is an Italian dairy product made from cow’s milk, typical of the Emilia Romagna region, that belongs to Stracchino-style cheeses, and is sold after a few days, packaged in simple, white, grease-proof paper, and is characterized by a white, shapeless, and grainy form with a delicate, sweet taste [39]. In addition, Squacquerone di Romagna is a protected designations of origin (PDO) product. Although it has regional importance, the few literature data show that it is characterized, similar to Crescenza cheese, by a low content of Na and an optimum Ca/P ratio compared to other cheeses [39]. More specifically, the impacts of the inclusion of *L. lactis* subsp. *lactis* CBM 21 in a novel starter system on the cheese safety, shelf-life, and quality in terms of proteolytic and lipolytic activities, sensory, and volatile molecular profiles were assessed, considering the traditional product as a control. 

## 2. Materials and Methods

### 2.1. Microbial Strain Conditions

The *Lactococcus lactis* subsp. *lactis* CBM 21, isolated from Mozzarella cheese [13], belongs to the Department of Biotechnology of Verona University. The strain was grown overnight (30 °C for 16 h) in M17 broth (Oxoid, Basingstoke, UK) under aerobic conditions. Microbial cultures were centrifuged at 8000× *g* for 20 min at 4 °C. The cell pellet was washed twice with physiological solution (0.9% NaCl in distilled water) and suspended in commercial whole milk. As a starter culture for Squacquerone cheese production, commercial *Streptococcus thermophilus* St 0.20 (Sacco S.R.L., Como, Italy) was used as a freeze-dried culture. The starter was added to milk following the producer guidelines.

### 2.2. Production of Squacquerone Cheese

The production of Squacquerone cheese was done in a pilot-scale plant of a local cheese factory (Mambelli, Bertinoro, Italy). Two batches of pasteurized whole cow’s milk (100 L) were warmed up to a temperature of 42 °C. The starter *S. thermophilus* St 0.20 was inoculated at 6.0 Log Colony-forming unit (CFU)/ml. One batch was also inoculated with *L. lactis* subsp. *lactis* CBM 21 at 7.0 Log CFU/mL. Forty min later, NaCl (0.7%) and 37 mL of rennet (12,000 international milk coagulating units (IMCU)/mL, 80% chymosin, and 20% pepsin, Bellucci Modena, Italy) were added to both batches. After coagulation, approximatively 20 min later, the curd was cut and moved to traditional baskets. The products were allowed to rest until a pH of 5.15 was reached and were then placed at 4 °C. The next day, the cheeses were packed under a modified atmosphere and stored at 4 °C up to 15 days. Three different types of cheese production on different days were performed for each cheese typology.

### 2.3. Microbiological Analyses and pH

Twenty g of cheese was added to 180 ml of sterile sodium citrate solution (20 g/L) and homogenized for 3 min by a stomacher (BagMixer 400P, Interscience, Saint-Nom-la-Bretèche, France). Serial dilutions of the homogenized samples were done in a physiological solution (0.9% NaCl), and aliquots of each dilution were spread onto the surface of different selective agar media. The enumeration of *Streptococcus thermophilus* was done on M17 agar (Oxoid, Basingstoke, Hampshire, UK) (42 °C, 48 h), while Lactococci were counted on M17 agar after 48 h at 30 °C. Moreover, the presence of the biocontrol agent was assured by checking the bacterial colony morphology and through molecular identification of the colonies by sequencing the 16S rRNA region according De Angelis et al. [40]. Total coliforms and yeasts were detected on Violet Red Bile agar (VRBA, Oxoid, Basingstoke, Hampshire, UK) (37 °C, 24 h) and Yeast extract Peptone Dextrose agar (YPD, Oxoid, Basingstoke, Hampshire, UK) (25 °C, 48 h), respectively. The presence of pathogenic species such as *Listeria monocytogenes*, *Salmonella enteritidis*, and *Escherichia coli* was monitored during the refrigerated storage of cheeses according to the ISO methods 11290, 6579, and 16649, respectively. pH was analyzed in cheese samples diluted in distilled water at a ratio 1:1 by a pH-meter (PH BASIC 20, Crison, Hach Lange, Italy).

### 2.4. Proteolysis, Lipolysis, Volatile Profiles

Proteolysis was examined by using sodium dodecyl sulfate–polyacrylamide gel electrophoresis (SDS-PAGE, Bio-Rad Laboratories, Hercules, US). The method proposed by Kuchroo and Fox [41] was used for protein extraction, while the running conditions were set up according to Tofalo et al. [42]. The lipid extraction from cheese samples and the analysis of the concentration of free fatty acids (FFAs) were done according to Vannini et al. [43]. The volatile molecular profiles of the cheese samples were detected by the solid-phase microextraction combined to chromatography-mass spectrometry (GC/MS/SPME) technique following the methodology reported by Burns et al. [44]. The identification of the molecules was performed by using the mass spectra database from National Institute of Standards and Technology (NIST version 2011, Gaithersburg, US).

### 2.5. Textural Profile Analyses

The sample texture was determined after 1, 4, 6, and 13 days of refrigerated storage by a texture analyzer (TA-DHI, Stable MicroSystem, Godalming, UK) following the protocol reported by Patrignani et al. [45].

### 2.6. Sensory Evaluation

The sensory evaluation was performed after the refrigerated storage of the cheeses. The analysis was done by 25 trained panelists on 20 g of each cheese sample following the recommendations of Standard 8589 (ISO, 1988), as reported by Gallardo-Escamilla et al. [46]. The parameters evaluated were flavor, creaminess, color, bitter off-flavors, and overall acceptance. For each parameter, the grading scale ranged between 0 (low or poor) and 5 (high or excellent).

### 2.7. Statistical Analysis

The data are the mean of three biological replicates. The obtained data were analyzed by Statistica software (version 8.0; StatSoft, Tulsa, Oklahoma, USA). The significant differences among samples at the same storage time were detected using ANOVA followed by the LSD test at the *p* < 0.05 level. Principal component analysis (PCA) was performed on the volatile molecular profiles of the cheese samples by Statistica software (version 8.0; StatSoft, Tulsa, Oklahoma, USA).

## 3. Results

### 3.1. Microbiological Analyses and pH

Cheeses produced with the two starter systems (i.e., only *S. thermophilus* St 0.20 and *S. thermophilus* St 0.20 plus *L. lactis* subsp. *lactis* CBM 21, thereafter named traditional and innovative cheeses) were analyzed after 1, 4, 6, 8, 13, and 15 days of storage. The recorded pH values are presented in Table 1. The addition of *L. lactis* subsp. *lactis* CBM 21 did not modify the pH of the innovative cheese compared to the traditional one. In fact, one day after cheesemaking, the pH value of both the cheeses was 5.61 ± 0.13, while at the end of refrigerated storage, it was 5.38 ± 0.02 and 5.41 ± 0.02 for traditional and innovative cheeses, respectively. From a microbiological point of view, the presence of *L. monocytogenes*, *Salmonella* spp., and *E. coli* was verified at each time of sampling in both the cheese typologies. These pathogenic bacteria were not found during storage (data not shown). The microbiological results concerning *S. thermophilus*, *L. lactis*, total coliform, and yeast cell loads are reported in Table 2 and Table 3. The highest yeasts load were attained in the traditional cheese (1.4 Log CFU/g) after four days of refrigerated storage. However, at the end of the shelf-life (15 days), 1.0 Log CFU/g of yeasts were found in both cheese types. The total coliforms never exceeded 1.2 Log CFU/g in the innovative cheese during the shelf-life while they reached the highest values of 2.4 Log CFU/g in the traditional cheese after four days storage. However, in control cheese, after 15 days of storage, a coliform load of 1.7 Log CFU/g was detected.

The *S. thermophilus* starter attained cell loads of 7.1 and 7.3 Log CFU/g in the innovative and control cheese, respectively, after 1 day of refrigerated storage. In the traditional cheese, *S. thermophilus* remained quite stable for the considered period, while it increased during the refrigerated storage (up to 11 days of storage) in the innovative cheese. After 15 days, it attained a level of 7.2 Log CFU/g in the innovative cheese. The cell load of *L. lactis* subsp. *lactis* CMB 21 in the innovative cheese increased during the storage, reaching levels of 8.1 Log CFU/g after 15 days.

### 3.2. Lipolysis, Proteolysis, and Volatile Profile 

The two types of cheeses were analyzed for their lipolytic, proteolytic, and volatile molecular profiles during the shelf-life. The lipolytic patterns were investigated after 1, 6, 11, and 15 days of storage, and the data obtained for free fatty acids (FFAs) are reported in Table 4. After 1 day of storage, the most released FFAs were C16:0, C18:0, C14:0, and C18:1 in both cheese types. The amount of FFAs increased over time for both cheeses. After 15 days, C18:1, which is considered a precursor for many aroma compounds, was significantly (*p* < 0.05) more abundant in the innovative cheese. The total amount of FFAs after 15 days of storage was significantly (*p* < 0.05) higher for the innovative cheese (373 ppm) than for the traditional one (291 ppm).

Figure 1a–c show the proteolytic patterns by sodium dodecyl sulfate–polyacrylamide gel electrophoresis (SDS-page) of the two cheese types after 1, 4 (Figure 1a), 6, 8, (Figure 1b), 11, and 15 (Figure 1c) days of storage. It is clear that the cheese proteolytic profiles were not significantly influenced by the adjunct of CBM 21. After four days of storage, the bands corresponding to 50 Kilodalton (KDa) disappeared, compared to those of 1 day, while the bands with a molecular weight between 25 and 15 KDa increased in both the cheeses, regardless of the presence of CBM 21.

The volatile compounds of the cheeses, at different times of storage, were investigated by GC/MS coupled with solid phase micro extraction (SPME). This approach permitted us to detect 35 compounds belonging to different chemical classes. The main detected molecules expressed as a relative percentage are reported in Table 5. After 1 day of storage, 2-butanone and 2-3 butanedione were detected in similar amounts in both the traditional and innovative cheeses, while 3 hydroxy-2 butanone was present at the highest level in the innovative cheese. During storage, a common trend observed, regardless of the type of sample, was the increase in acid and alcohol relative percentages associated with a decrease in esters and ketones. However, innovative cheeses were characterized by the highest amounts of acids, mainly short chain fatty acids such as butanoic, hexanoic, and octanoic acids, while in traditional cheeses, the highest increase in alcohol, mainly ethanol, was detected. In this framework, principal component analysis (PCA) in relation to the storage time was performed in order to better understand the effect of *L. lactis* subsp. *lactis* CBM 21 on the volatile molecular profile of Squacquerone cheese. The projections of the cases and the variables in the factorial plane determined by principal component 1 (PC1) and principal component 2 (PC2) are shown in Figure 2. It is evident that the adjunct culture significantly affected the volatile molecular profile of Squacquerone at the beginning of the storage (1 day). In fact, the innovative cheese, after 1 day of refrigerated storage, was well separated along the PC2, which explained 29.81% of the variance among the samples, and, to a less extent, along PC1, which explained 34.99% of the variance. The most significant molecules responsible for the differences between the control and innovative cheeses were mainly ketones (i.e., 2-butanone, 3 hydroxy-2- butanone, acetone, 2,3- butanedione), acids (butanoic, hexanoic, and octanoic acids), and esters (acetic acid ethenyl ester and ethyl acetate). During the storage, the effects of the adjunct culture decreased. In fact, the two cheese typologies were still separated along PC1 and PC2 after 11 days, while, at the end of the refrigerated storage, they grouped together. After 15 days, both cheeses were characterized principally by 2-butanone and in lower amounts, by esters and alcohols.

### 3.3. Texture Analyses 

The texture analyses showed initial significant differences in hardness and consistency between the two different cheeses (Figure 3). In fact, at the beginning, the innovative cheeses were characterized by lower values of hardness and consistency compared to the control cheeses. However, these differences decreased over time, and no significant differences were found between the two cheese typologies after 15 days of refrigerated storage.

### 3.4. Sensory Analysis

The panel test, performed after 6 and 15 days of storage on the two different cheese typologies, indicated that the innovative cheese was preferred by the evaluators compared to the traditional one for all the attributes taken into consideration (Figure 4). Indeed, after six days, the innovative cheese had better scores compared to the traditional one for the overall acceptance, taste, creaminess, and flavor, while it received a lower score for the bitter after-taste. After 15 days of storage, the innovative cheese received a positive evaluation for all the attributes although the differences compared to traditional Squacquerone were reduced.

## 4. Discussion

Our results indicate that the use of a selected strain of *L*. *lactis* subsp. *lactis*, which produces nisin Z, is suitable for the production of a high-quality cheese, endowed with a longer shelf-life, characterized by specific ripening profiles and improved safety and sensory features. Indeed, the microbiological data showed a significant reduction of the microorganisms (such as yeasts and *Pseudomonas* spp.) involved in the spoilage of both soft and hard cheeses [47,48,49,50]. In addition, total coliforms were totally inhibited in samples with the bacteriocin producer strain. Data obtained confirm the effectiveness of *L. lactis* as a biocontrol agent against Gram-positive and Gram-negative bacteria, thus improving the safety and shelf-life of dairy products [51,52]. In fact, the ability of nisin-producer strains to release this bacteriocin in dairy products during refrigerated storage has been documented [53]. In this framework, the suitability of the selected strain to be employed as an adjunct culture in Squacquerone cheese was demonstrated by its ability to reach a final cell load of 8.1 Log CFU/g at the end of storage, without negative effects on the viability of the conventional starter. Furthermore, the *S. thermophilus* starter strain was positively affected by the presence of CBM 21 in the early storage period. This stimulating effect on starter vitality can be ascribed to the higher early availability of FFAs, such as C18:1, which is an essential factor for the growth of lactic acid bacteria [44,54]. The addition of CBM 21 as an adjunct culture allowed a higher and precocious release of FFAs in cheese during the refrigerated storage. These data are in agreement with those of Ávila et al. [55], who attributed the precocious accumulation of FFAs in Hispanico cheese to the release of intracellular esterases by a nisin-producer strain, *L. lactis* subsp. *lactis* (INIA 415). Moreover, Ávila et al. [56] also observed an increased level of FFAs in semi-hard cheese inoculated with *L. lactis* subsp. *lactis* INIA 63. The authors attributed the increased lipolysis to the lysis of the starter culture (belonging to the species *Lactobacillus helveticus*) mediated by the bacteriocin and the subsequent release of its esterases. On the other hand, it is widely reported that lactic acid bacteria have a minor role in the lipolytic pattern of dairy products compared to the proteolytic one and their contribution is mainly in terms of esterase activity rather than the production of lipases [57,58]. On the other hand, the precocious release of FFAs can be responsible for the positive sensory evaluation of the innovative cheese. In fact, FFAs can directly contribute to the flavor and taste of dairy products (mainly short FFAs, responsible for the piquant notes) or can affect them as precursors of key aroma compounds [59]. By contrast, the strain *L*. *lactis* subsp. *lactis* CBM 21, used in the present research, did not contribute to the proteolytic patterns of the innovative cheese in a significant manner, although the positive contribution of such species to the ripening of several cheeses, due to the well-recognized protease and peptidase activities of this strain, is well known. This can be ascribed to the short time of storage and the low temperature adopted during the ripening. Indeed, it is well known that cheese proteolysis is affected by ripening time and temperature [60,61]. The increase in temperature has long been used to accelerate proteolysis and consequently to reduce the cheese ripening time [62,63]. The addition of *L*. *lactis* subsp. *lactis* CBM 21 resulted in a different cheese volatile molecular profile. At the beginning of the refrigerated storage, the innovative cheese was characterized by the highest amount of 2,3 butanedione (diacetyl) and 2-butanone,3-hydroxy (acetoin). These aroma compounds, together with acetaldehyde and 2,3 butanediol, are generally regarded as the principal aroma compounds of yoghurt and soft-cheese-like products [64,65]. In particular, the diacetyl, derived from citrate metabolism, plays a crucial role in several dairy products affecting the creamy and buttery flavor, and its presence is characteristic of Camembert, Cheddar, and Emmental cheeses [66,67], as well as of soft cheeses [68]. The ability of *L*. *lactis* to produce these key volatile compounds is well known, and this is one of the main reasons why it is exploited at the industrial level [8,69]. The differences analytically detected between the two types of cheeses were perceived by the panelists involved in this research: the innovative cheese was the most appreciated for the entire considered storage time, which resulted in a creamier flavor, particularly after 4–6 days. These data are in agreement with the texture analysis, since the cheese produced by adding CBM 21 was characterized by lower hardness and consistency. Although proteolysis, which plays a crucial role in the texture development [70], was not significantly different between the two types of Squacquerone, the cheese texture can be affected by other factors, such as cheese composition, processing conditions, and the ripening process [63].

However, the difference detected both by GC/MS-SPME and the panel test, as well as the texture differences decreased over time were reduced at the end of the shelf-life.

The results of this study indicate that the use of the strain *L. lactis* CBM 21 is suitable to obtain a high-quality Squacquerone cheese. Although PDO production imposes precise protocols to obtain safe products, the use of *Lactococcus lactis* CBM 21 could represent a further strategy to improve the safety of Squacquerone cheese, imparting good and innovative quality features. In this view, in a saturated market such as the dairy one, the possibility to obtain a product with improved quality characteristics and better safety may represent a good strategy for product differentiation.

## 5. Conclusions

The present study revealed that the nisin Z-producer *L. lactis* subsp. *lactis* CBM 21 was suitable for the production of Squacquerone cheese. In fact, its addition as a biocontrol agent had positive effects not only on the safety, but also on the overall quality of the cheese, giving rise to a product potentially appreciated by the consumers for its specific cheese volatile and textural profiles. Therefore, the use of a starter system which includes strains with superior technological and safety attributes could contribute to enlarge the gamut of high-quality dairy products, improving their safety without having any detrimental effects on the sensory characteristics.

## Figures and Tables

**Figure 1 animals-10-00782-f001:**
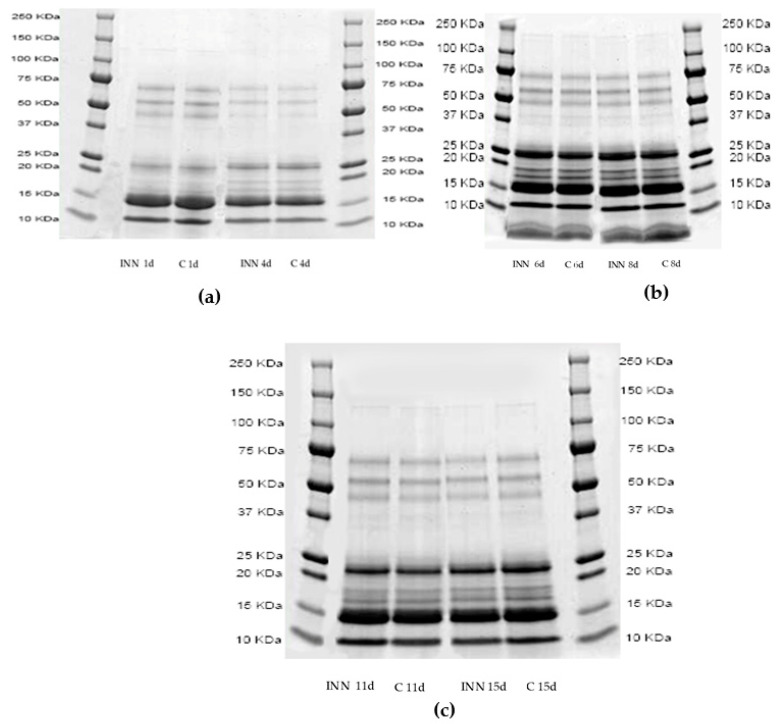
Sodium dodecyl sulfate–polyacrylamide gel electrophoresis profiles of pH 4.6 soluble fractions of traditional (C) and innovative (INN) Squacquerone cheeses after 1 and 4 (**a**), 6 and 8 (**b**), and 11 and 15 (**c**) days.

**Figure 2 animals-10-00782-f002:**
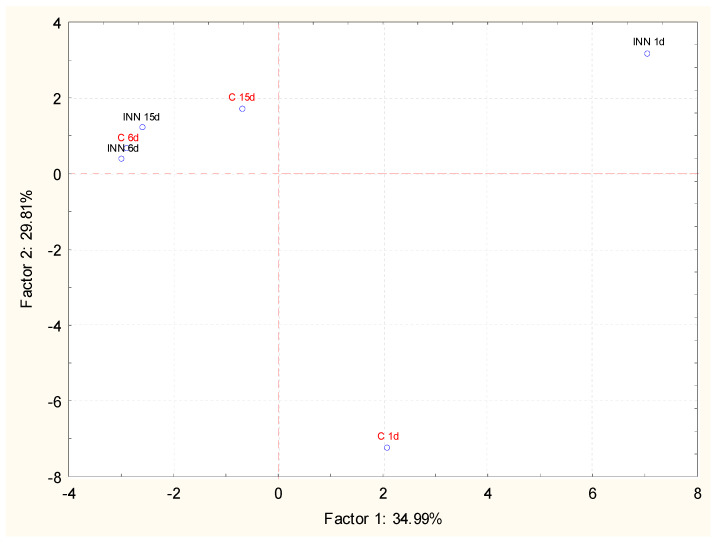
Projection of the different Squacquerone cheeses (INN and C) on the factor plane (1-2) at different times of storage (1, 6, and 15 days) on the basis of the volatile profiles detected by solid-phase microextraction combined to chromatography-mass spectrometry technique (GC/MS SPME).

**Figure 3 animals-10-00782-f003:**
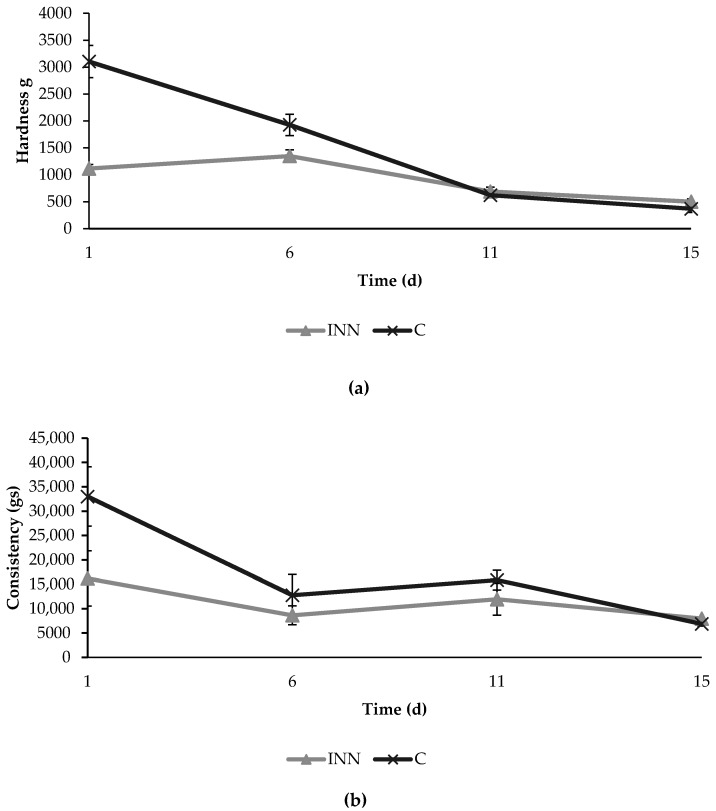
Texture parameters in terms of hardness (**a**) and consistency (**b**) detected in traditional (C) and innovative (INN) Squacquerone cheeses during the refrigerated storage.

**Figure 4 animals-10-00782-f004:**
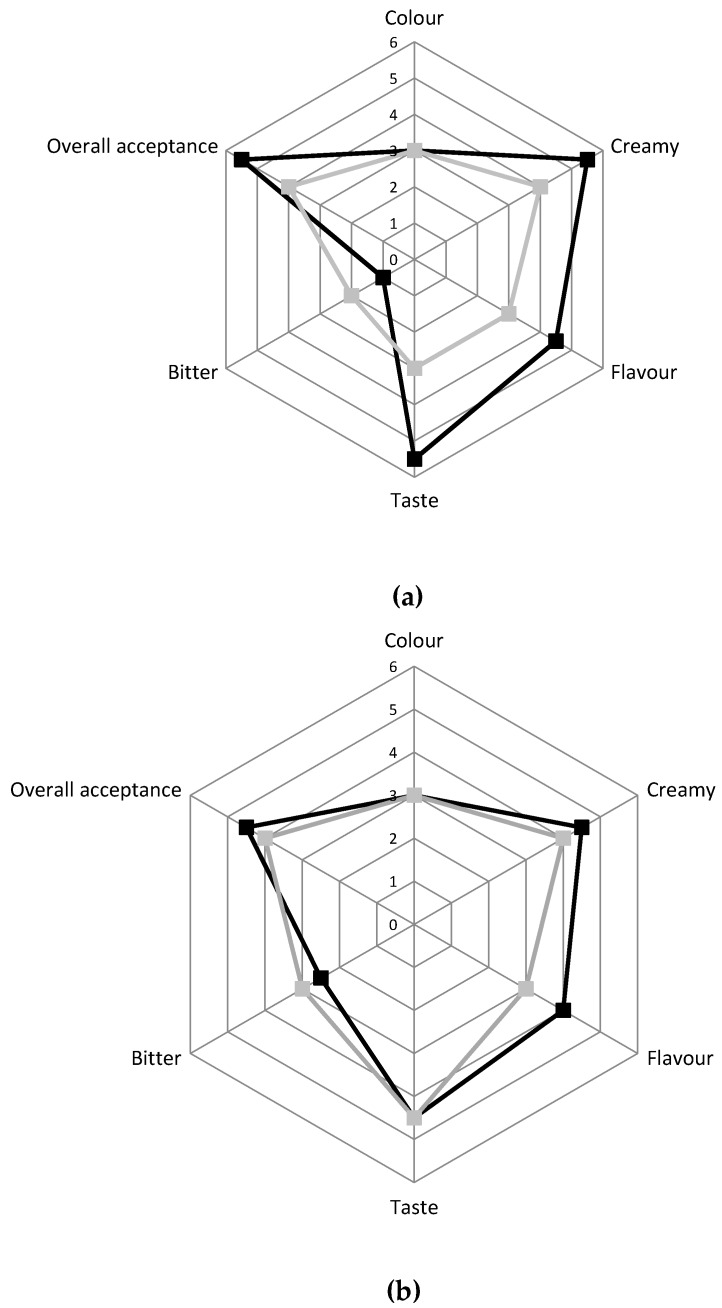
Sensory evaluation of C ■ and INN ■ Squacquerone cheeses performed after 6 days (**a**) and 15 days (**b**) of refrigerated storage.

**Table 1 animals-10-00782-t001:** pH evolution in innovative and traditional Squacquerone cheeses during refrigerated storage.

Time (days)	INN ^1^	C ^2^
1	5.60 ± 0.02	5.60 ± 0.01
6	5.46 ± 0.13	5.43 ± 0.11
8	5.57 ± 0.03	5.62 ± 0.01
11	5.30 ± 0.12	5.28 ± 0.04
15	5.42 ± 0.01	5.39 ± 0.02

^1:^ Innovative Squacquerone produced with a starter culture composed of *Streptococcus thermophilus* St 0.20 and *Lactococcus lactis* subsp. *lactis* CBM 21; ^2:^ Traditional Squacquerone produced with a starter composed of *Streptococcus thermophilus* St 0.20.

**Table 2 animals-10-00782-t002:** Cell load (Log CFU/g) of *Streptococcus thermophilus* and *Lactococcus lactis* CBM 21 in innovative (INN) and traditional (C) Squacquerone cheeses during refrigerated storage. At the same time of storage, the means ± SD of *S. thermophilus* followed by different superscript letters (^a^ or ^b^) are significantly different, *p* < 0.05.

Time (days)	INN ^1^		C ^2^
	*S. thermophilus*	*L. lactis CBM 21*	*S. thermophilus*
	Log CFU/g	Log CFU/g	Log CFU/g
1	7.6 ± 0.3 ^a^	7.0 ± 0.5	7.5 ± 0.2 ^a^
6	8.1 ± 0.3 ^a^	7.6 ± 0.4	7.6 ± 0.2 ^a^
8	8.0 ± 0.3 ^a^	7.9 ± 0.2	7.1 ± 0.3 ^b^
11	8.1 ± 0.2 ^a^	8.2 ± 0.3	7.3 ± 0.2 ^b^
15	7.3 ± 0.3 ^a^	8.0 ± 0.4	6.7 ± 0.3 ^a^

^1:^ Innovative Squacquerone produced with a starter culture composed of *Streptococcus thermophilus* St 0.20 and *Lactococcus lactis* subsp. *lactis* CBM 21; ^2:^ Traditional Squacquerone produced with a starter composed of *Streptococcus thermophilus* St 0.20.

**Table 3 animals-10-00782-t003:** Cell load (Log CFU/g) of yeasts and total coliforms, in innovative and traditional Squacquerone cheeses during refrigerated storage. At the same time of storage, means ± SD followed by different superscript letters (^a–c^) are significantly different, *p* < 0.05.

Time (days)	INN ^1^		C ^2^	
	Yeasts	Total Coliforms	Yeasts	Total Coliforms
	Log CFU/g	Log CFU/g	Log CFU/g	Log CFU/g
1	- *	1.2 ± 0.2 ^a^	1.3 ± 0.2 ^a^	2.3 ± 0.2 ^b^
6	- *	- *	1.8 ± 0.3 ^a^	2.4 ± 0.3 ^a^
8	1.0 ± 0.1 ^a^	1.0 ± 0.1 ^a^	1.9 ± 0.1 ^b^	1.9 ± 0.2 ^b^
11	1.2 ± 0.2 ^a^	- *	2.2 ± 0.2 ^b^	2.1 ± 0.3 ^b^
15	- *	1.0 ± 0.1 ^a^	2.8 ± 0.2 ^c^	2.0 ± 0.1 ^b^

^1:^ Innovative Squacquerone produced with a starter culture composed of *Streptococcus thermophilus* St 0.20 and *Lactococcus lactis* subsp. *lactis* CBM 21; ^2:^ Traditional Squacquerone produced with a starter composed of *Streptococcus thermophilus* St 0.20; *: under the detection limit (0.5 Log CFU/g).

**Table 4 animals-10-00782-t004:** Free fatty acid (FFA)concentration (ppm) in innovative and traditional Squacquerone cheeses after 1, 6, 11, and 15 days of refrigerated storage. In the same line, for each FFA, means ± SD followed by different superscript letters (^a–f^) are significantly different, *p* < 0.05.

FFA	INN ^1^				C ^2^			
	1d	6d	11d	15d	1d	6d	11d	15d
C14:0	2.0 ± 0.2 ^a^	1.9 ± 0.2 ^a^	3.3 ± 0.3 ^b^	3.0 ± 0.4 ^b^	1.9 ± 0.4 ^a^	2.3 ± 0.2 ^a^	3.4 ± 0.4 ^b^	4.0 ± 0.6 ^b^
C16:0	44.2 ± 2.3 ^a^	93.4 ± 4.5 ^c^	152.0 ± 7.8 ^e^	183.0 ± 9.8 ^f^	45.0 ± 1.9 ^a^	73.5 ± 6.7 ^b^	123.1 ± 9.3 ^d^	185.0 ± 10.5 ^f^
a-C17:0	-*	-	-	1.5 ± 0.2	-	-	-	-
C17:0	-	-	-	3.5 ± 0.5	-	-	-	-
C18:2	-	-	-	-	-	-	1.1 ± 0.1	-
(C18:1 d9)	2.0 ± 0.2 ^a^	2.3 ± 0.2 ^a^	4.2 ± 0.4 ^b^	7.6 ± 0.7 ^d^	1.5 ± 0.3 ^a^	2.1 ± 0.4 ^a^	4.3 ± 0.6 ^b^	6.0 ± 0.4 ^c^
C18:0	30.0 ± 1.5 ^a^	27.0 ± 2.3 ^a^	82.1 ± 9.8 ^c^	172.7 ±1 2.4 ^d^	28.0 ± 1.7 ^a^	42.0 ± 3.9 ^b^	96.3 ± 8.7 ^c^	96.0 ± 10.7 ^c^
C20:0	-	-	-	2.0 ± 0.3	-	-	-	-
total FFAs	78.2 ± 4.1 ^a^	124.6 ± 8.1 ^b^	241.6 ± 18.2 ^c^	373.3 ±2 4.9 ^e^	76.4 ± 4.3 ^a^	119.9 ± 11.1 ^b^	228. 2 ± 19.2 ^c^	291.0 ± 22.1 ^d^

^1:^ Innovative Squacquerone produced with a starter culture composed of *Streptococcus thermophilus* St 0.20 and *Lactococcus lactis* subsp. *lactis* CBM 21; ^2:^ Traditional Squacquerone produced with a starter composed of *Streptococcus thermophilus* St 0.20; *: under the detection limit.

**Table 5 animals-10-00782-t005:** Volatile compounds (expressed as relative percentages) detected through the GC-MS-SPME technique in innovative (INN) and traditional (C) Squacquerone cheeses after 1, 6, 11, and 15 days of storage. In the same line, relative to each FFA, means ± SD followed by different superscript letters are significantly different, *p* < 0.05.

Volatile Molecule	INN ^1^				C ^2^			
	1d	6d	11d	15d	1d	6d	11d	15d
1-Pentene, 2,4,4-trimethyl-	1.63	0.35	3.36	2.71	5.09	0.59	1.70	5.31
Thiophene	0.00	0.57	0.29	0.23	3.56	1.09	1.16	0.00
TOTAL HYDROCARBONS	1.63	0.92	3.65	2.93	8.65	1.68	2.86	5.31
Acetone	7.26	2.21	3.84	4.81	3.20	2.10	4.02	6.92
2-Butanone	4.45	6.93	4.71	7.26	4.83	9.97	4.99	2.73
2,3-Butanedione	13.87	12.10	14.33	10.14	13.86	11.63	17.61	13.33
2,3-Pentanedione	6.38	7.61	6.10	4.93	3.22	5.08	1.87	5.11
3-Hexanone	6.81	0.72	0.62	0.00	9.48	1.05	0.00	0.61
Hexanone	2.32	1.15	0.46	0.00	0.52	0.76	0.00	0.00
3-Pentanone, 2-methyl-	0.00	2.99	0.62	0.28	1.38	0.88	0.38	0.55
2-Pentanone, 3-methyl-	0.00	2.33	0.18	0.42	1.04	1.50	0.17	0.54
2-Butanone, 3-hydroxy-	16.29	13.44	16.36	9.03	7.98	12.82	7.95	12.24
Cyclohexanone,2-chloro-2-methyl-	9.79	0.00	3.25	1.12	0.72	0.00	2.08	2.87
TOTAL KETONES	67.17	49.47	50.48	37.99	46.23	45.80	39.08	44.90
Ethyl acetate	0.00	1.60	2.10	7.43	0.00	1.34	3.09	1.91
Propanoic acid, ethyl ester	0.00	0.00	0.23	0.23	0.00	0.00	0.19	0.33
Acetic acid ethenyl ester	11.01	2.37	1.79	0.28	19.59	2.04	0.20	1.17
Butanoic acid, ethyl ester	0.00	0.86	0.42	4.54	0.26	1.37	3.40	0.63
Pentanoic acid, ethyl ester	0.00	0.00	0.00	1.65	1.19	0.00	0.73	0.32
Pentanoic acid, 4-methyl-, ethyl ester	0.00	0.00	0.00	0.23	0.00	0.00	0.00	0.00
TOTAL ESTERS	11.01	4.84	4.54	14.35	21.04	4.76	7.62	4.35
Ethanol	2.06	20.68	17.00	16.76	6.42	30.31	35.14	22.62
1-Propanol, 2-methyl-	0.15	0.76	0.25	2.40	0.67	3.00	1.95	0.50
Cyclopropanemethanol, (1-methylethyl)-	1.04	0.94	2.83	2.00	0.56	0.85	1.31	2.88
2-Hexanol, 2,3-dimethyl-	0.14	0.66	1.15	1.52	4.11	0.65	0.58	1.34
1-Butanol, 3-methyl-	1.17	1.46	2.93	6.76	1.60	1.20	2.92	1.02
3-Pentanol, 2-methyl-	1.86	0.48	3.00	0.00	1.00	0.00	0.75	1.68
2-Octanol,	2.07	0.00	2.42	0.00	0.92	2.13	1.81	1.15
Phenylethyl Alcohol	0.54	0.00	0.00	2.41	0.09	0.00	0.11	2.08
TOTAL ALCOHOLS	9.04	24.98	29.57	31.86	15.37	38.14	44.56	33.28
Benzaldehyde	1.98	0.00	0.00	1.19	1.54	0.00	1.00	2.21
TOTAL ALDEHYDES	1.98	0.00	0.00	1.19	1.54	0.00	1.00	2.21
Butanoic acid	0.00	9.78	3.01	3.65	0.00	2.70	0.00	2.78
hexanoic acid	0.68	4.16	3.51	3.01	1.01	2.95	1.49	2.33
Heptanoic acid	0.16	0.65	0.00	0.76	0.66	0.00	0.13	0.59
Octanoic Acid	7.44	5.20	4.22	3.64	4.27	2.55	2.26	3.11
TOTAL ACIDS	8.28	19.78	10.74	11.06	5.94	8.20	3.89	8.81
TOTAL MOLECULES	99.11	100.00	98.99	99.40	98.76	98.57	99.00	98.86

^1:^ Innovative Squacquerone produced with a starter culture composed of *Streptococcus thermophilus* St 0.20 and *Lactococcus lactis* subsp. *lactis* CBM 21; ^2:^ Traditional Squacquerone produced with a starter composed of *Streptococcus thermophilus* St 0.20; The coefficients of variation, expressed as the percentage ratios between the standard deviations and the mean values, ranged between 2% and 6%.

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
