# Peer review of "Suitability of the Nisin Z-producer Lactococcus lactis subsp. lactis CBM 21 to be Used as an Adjunct Culture for Squacquerone Cheese Production"

_animals, 2020, doi:10.3390/ani10050782_

Round 1
Reviewer 1 Report
The manuscript describes the technological and safety aspects of a nisin Z-producer strain of Lactotoccus lactis strain as an adjunct culture for Squacquerone cheese production in a pilot-scale plant. The manuscript is well-written and the topic is of interest to the international scientific community.
Below some specific comments:
Title: CBM21 should be CBM 21
Abstract:
Line 16: substitute "de" with "the"
Keywords: substitute "Lactis with "lactis"
Introduction
Line 35: check the two times "dairy", it is a repetition.
Line 39: "dramatically" has a negative meaning in this sentence, change with an appropriate term.
Lines 58-59: please check and eventually correct or specify the statement that culture-independent analytical methods showed the presence of alive cells.
Materials and Methods
Line 92: please specify how many replicates of the Squaquerone cheese production have been performed.
Line 125:"the three repetitions" are referred to biological or technical replicates?
Tables 2 and 3: the Anova letters are missed. Please add the meaning of "-" within table legends. Check the bold letters of the names within the Tables.
Results
Line 191: add appropriately the panel a, b, c within the text. In Figure 1 it is difficult to compare the intensity of the bands due to the different backgrounds of the gels in panel a, b, c. Is it possible to clarify the background of gels b and c?
Discussion
Lines 311-312: check the english, you should probably eliminate the word "and" before "were".
References
Check the reference Cardinali et al. 2016. The correct one should be:
Yeast, 33 (8) pp. 403-414.
Author Response
REVIEWER 1
The manuscript describes the technological and safety aspects of a nisin Z-producer strain of Lactotoccus lactis strain as an adjunct culture for Squacquerone cheese production in a pilot-scale plant. The manuscript is well-written and the topic is of interest to the international scientific community.
Thanks to the reviewer for the comments
Below some specific comments:
Title: CBM21 should be CBM 21
Done
Abstract:
Line 16: substitute "de" with "the"
Done
Keywords: substitute "Lactis with "lactis"
Done
Introduction
Line 35: check the two times "dairy", it is a repetition.
Done
Line 39: "dramatically" has a negative meaning in this sentence, change with an appropriate term.
We replace the word “dramatically” with “markedly”
Lines 58-59: please check and eventually correct or specify the statement that culture-independent analytical methods showed the presence of alive cells.
We revised the sentence to better clarify the statement
Materials and Methods
Line 92: please specify how many replicates of the Squaquerone cheese production have been performed.
We specify that 3 different cheese productions were performed for each sample typology.
Line 125:"the three repetitions" are referred to biological or technical replicates?
We specified in Material and Method section that the statistical analysis is referred to three biological replicates
Tables 2 and 3: the Anova letters are missed. Please add the meaning of "-" within table legends. Check the bold letters of the names within the Tables.
Thanks to the reviewer for the suggestions, we introduced the ANOVA in table 2 (only for S.thermophilus cell load since CBM21 cell load was referred to only innovative cheese) and table 3. We explained that the meaning of “-“is under the detection limit. We revised the presence of bold letters in tables
Results
Line 191: add appropriately the panel a, b, c within the text. In Figure 1 it is difficult to compare the intensity of the bands due to the different backgrounds of the gels in panel a, b, c. Is it possible to clarify the background of gels b and c?
Thanks to the reviewer for the suggestion, we introduced the appropriate panel within the text and we reduced the background of gel b and c
Discussion
Lines 311-312: check the english, you should probably eliminate the word "and" before "were".
Done
References
Check the reference Cardinali et al. 2016. The correct one should be:
Yeast, 33 (8) pp. 403-414.
We revised the reference Cardinali et al. 2016 as suggested

Reviewer 2 Report
In general the manuscript is well written and scientifically well structured.
Few misprints need attention:
Lines 16-17: “The finding of this research indicates that de the tested Lactococcus lactis stain [strain] representS a suitable…”
Lines 33-35: please fragment the sentence, because it is very long without any punctuation
Line 89: please specify the measure units of the liquid rennet 12,000 U: U/mg? or IMCU?
Line 94: “Twenty g of cheese was placed…” WERE instead of WAS
Line 191 and 211: “Figure” instead of “Figures”
Line 312: take off final double point.
Moreover:
- Along the whole text and tables and figures it would be recommended to indicate the experimental group with the same name or initials; so please make uniform the name of the innovative Squacquerone cheese, choosing between Innovative, IN, INN C, and the traditional Squacquerone between Traditional, Control, C C. This may lead the reader to confusion.
- Table 1: since the differences are not significant, simply cancel the “a” superscript letters and the note in the legenda.
- Figure 2: add further details to the caption
- As far as the introduction, as the authors well know, Squacquerone cheese is a PDO, that is more than a traditional or typical cheese. The PDO cheese has specific characteristics concerning healthy, safety, shelf-life and organoleptic profile. The manuscript presents an interesting proposal of innovation, however the mention of the traditional PDO Squacquerone cheese is due, together with the inclusion of the issue onto the discussion, where the impression is that the traditional product is less safe than innovative one.
- As far as the Methods, further details on the milk (line 86 - raw whole milk?) and its microbiological quality would be appreciated, mostly considering the presence of Coliform population in both the cheeses.
- The sentence (lines 315-317) “In fact, its addition as a biocontrol agent had positive effects not only on the safety, but also on the overall quality of the cheese, giving rise to a product appreciated by the consumers for its specific cheese volatile and textural profiles” assumes that a consumer test was carried out, while the overall acceptability was expressed by the trained panelist. In this case I would suggest to add “…potentially appreciated…”.
At last, the volatile compounds detected in the cheeses deserve to be listed into a table, being fig. 2 (b) very interesting but a bit arduous to read out of the PCA.
Author Response
REVIEWER 2
In general the manuscript is well written and scientifically well structured.
Thanks to the reviewer
Few misprints need attention:
Lines 16-17: “The finding of this research indicates that de the tested Lactococcus lactis stain [strain] representS a suitable…”
Done
Lines 33-35: please fragment the sentence, because it is very long without any punctuation
Done
Line 89: please specify the measure units of the liquid rennet 12,000 IMCU?
The measure UNIT was IMCU/ml
Line 94: “Twenty g of cheese was placed…” WERE instead of WAS
Done
Line 191 and 211: “Figure” instead of “Figures”
Done
Line 312: take off final double point.
Done
Moreover:
- Along the whole text and tables and figures it would be recommended to indicate the experimental group with the same name or initials; so please make uniform the name of the innovative Squacquerone cheese, choosing between Innovative, IN, INN C, and the traditional Squacquerone between Traditional, Control, C C. This may lead the reader to confusion.
Thanks to the reviewer for the suggestion, we revised the whole manuscript and INN and C as abbreviation of Innovative and traditional cheese
- Table 1: since the differences are not significant, simply cancel the “a” superscript letters and the note in the legenda.
Done
- Figure 2: add further details to the caption
As suggested, we added further information to the figure 2 caption.
- As far as the introduction, as the authors well know, Squacquerone cheese is a PDO, that is more than a traditional or typical cheese. The PDO cheese has specific characteristics concerning healthy, safety, shelf-life and organoleptic profile. The manuscript presents an interesting proposal of innovation, however the mention of the traditional PDO Squacquerone cheese is due, together with the inclusion of the issue onto the discussion, where the impression is that the traditional product is less safe than innovative one.
Following the reviewer suggestions, we add the concept of PDO both in the introduction (line 71-72) and in the discussion section (line 336-340).
- As far as the Methods, further details on the milk (line 86 - raw whole milk?) and its microbiological quality would be appreciated, mostly considering the presence of Coliform population in both the cheeses.
We added further information about the milk characteristics. Pasteurized whole milk was used, and coliforms were not detected in milk after the treatment. The presence in cheese samples after 1 day of storage can be an indicator of handling contamination.
- The sentence (lines 315-317) “In fact, its addition as a biocontrol agent had positive effects not only on the safety, but also on the overall quality of the cheese, giving rise to a product appreciated by the consumers for its specific cheese volatile and textural profiles” assumes that a consumer test was carried out, while the overall acceptability was expressed by the trained panelist. In this case I would suggest to add “…potentially appreciated…”.
Done
At last, the volatile compounds detected in the cheeses deserve to be listed into a table, being fig. 2 (b) very interesting but a bit arduous to read out of the PCA.
Following the reviewer suggestions, we replaced the figure 2b with table 5 that reports the relative percentages of the volatile molecules detected
